# Trends of Stunting Prevalence and Its Associated Factors among Nigerian Children Aged 0–59 Months Residing in the Northern Nigeria, 2008–2018

**DOI:** 10.3390/nu13124312

**Published:** 2021-11-29

**Authors:** Osita K. Ezeh, Tanvir Abir, Noor Raihani. Zainol, Abdullah Al Mamun, Abul H. Milton, Md. Rashidul Haque, Kingsley E. Agho

**Affiliations:** 1School of Health Sciences, Western Sydney University, Locked Bag 1797, Penrith, NSW 2750, Australia; t.abir73@gmail.com (T.A.); k.Agho@westernsydney.edu.au (K.E.A.); 2School of Business, Ahsanullah University of Science and Technology, Dhaka 1208, Bangladesh; 3Faculty of Entrepreneurship and Business, University Malaysia Kelantan, Kota Bharu 16100, Malaysia; raihani@umk.edu.my; 4Faculty of Business and Management, UCSI University, Kuala Lumpur 56000, Malaysia; amun7793@gmail.com; 5Epidemiology Resource Centre, Newcastle, NSW 2290, Australia; miltonhasnat@gmail.com; 6Department of Psychiatry, Sir Salimullah Medical College, Dhaka 1206, Bangladesh; rana.dmc@gmail.com; 7Translational Health Research Institute (THRI), Campbelltown Campus, Western Sydney University, Penrith, NSW 2571, Australia

**Keywords:** stunting in children, Nigerian northern geopolitical zones, children below 5 years, trends in stunting, prevalence of stunting, child’s stunting

## Abstract

Every year in Nigeria, malnutrition contributes to more than 33% of the deaths of children below 5 years, and these deaths mostly occur in the northern geopolitical zones (NGZs), where nearly 50% of all children below 5 years are stunted. This study examined the trends in the prevalence of stunting and its associated factors among children aged 0–23 months, 24–59 months and 0–59 months in the NGZs. The data of 33,682 recent live births in the NGZs, extracted from the Nigeria Demographic and Health Surveys from 2008 to 2018, were used to investigate the factors associated with stunting using multilevel logistic regression. Children aged 24–59 months reported the highest prevalence of stunting, with 53.3% (95% confidence interval: 52.0–54.6%). Multivariable analyses revealed four common factors that increased the odds of a child’s stunting across all age subgroups: poor households, geopolitical zone (northwest or northeast), being a male and maternal height (<145 cm). Interventional strategies focused on poverty mitigation through cash transfer and educating low socioeconomic mothers on the benefits of gender-neutral supplementary feeding and the timely monitoring of the offspring of short mothers would substantially reduce stunting across all age subgroups in the NGZs.

## 1. Introduction

Stunting is an indicator of chronic undernutrition in children below 5 years of age, and it remains a daunting public health and development challenges in Africa, including Nigeria. Over the last two decades, the number of stunted children below 5 years of age has decreased globally in all regions except Africa, where it increased by 12.9%, from 54.4 million in 2000 to 61.4 million in 2020 [1]. A stunted child is defined as one whose height is approximately the height-for-age below minus two standard deviations (−2 SD) from the median reference of the standard curve, as described by the World Health Organisation (WHO) Child Growth Standards [2]. Stunted growth is largely attributable to encumbered foetal growth due to maternal undernutrition and medical issues during pregnancy, socioeconomic constraints, recurrent child illness and inappropriate infant and young child feeding, resulting in the child’s inability to reach his/her cognitive and physical potential [3,4].

The prevalence of stunted children below 5 years of age in Nigerian remains very high, and it has hardly changed (i.e., 42.4% in 2003, 40.6% in 2008, 36.8% in 2013 and 36.8% in 2018) [5,6,7], indicating an uphill task in achieving the internationally agreed-upon target of a 40% decline in stunted children by 2025 [1]. Every year in Nigeria, more than one-third of all deaths among children below 5 years of age are attributed to malnutrition [8]. The burden of childhood undernutrition in Nigeria is very concerning, particularly in the northern geopolitical zones (NGZs), consisting of the north-central (NC), northeast (NE) and northwest (NW), where nearly 50% of all children below 5 years of age are stunted [9]. A recent report from a community-based management of acute malnutrition treatment and control centres established in 11 northern Nigerian states stated that severe wasting received more significant prominence in malnutrition treatment and care than stunting [10]. The inattention to stunted children could be one of the reasons why in the past decade and a half, the prevalence of stunted children in northern Nigeria has remained well above the 30% WHO prevalence threshold, indicating a range of public health concerns [11].

Considering the adverse effects of malnutrition, particularly on the most vulnerable groups, such as children below 5 years of age, the Nigerian government initiated and implemented a range of nutritional programmes and policies to effectively reduce child malnutrition. For example, the National Policy on Food and Nutrition in 2001 [12], the National Plan of Action on Food and Nutrition in 2004 [13] and the National Policy on Infant and Young Child Feeding in 2005 [14] whose core targets include a substantial reduction of childhood undernutrition. In addition, as part of the Child Development Grant Programme funded by the United Kingdom, a nutritional interventional programme was implemented in northern Nigeria between 2013 and 2019 [15]. Despite all these efforts, there has been little or no reduction in stunting among children below 5 years of age in Nigeria, particularly in the northern part of the country. For example, a recent report showed that 29%, 49% and 57% of children below 5 years of age in NC, NE and NW, respectively, were stunted or severely stunted between 2013 and 2018, as compared to their southern counterparts (18–25%) [7]. Therefore, designing effective nutritional intervention policies targeting children at heightened odds of chronic undernutrition is crucial and entails a clear delineation of factors associated with stunted children in the NGZs.

Few population-based or community-based studies in Nigeria have been conducted on the factors associated with stunting among children aged 0–23 months or 0–59 months. For instance, Akombi et al. [16] conducted a multilevel logistic regression analysis using standardised national representative data to investigate the factors related to stunting and severe stunting among children under 5 years of age in Nigeria. They suggested that being a male child, being a small or averaged-sized child, having a poor household, duration of breastfeeding >12 months, geopolitical zone (NC, NE and NW), and children who had diarrhoea two weeks prior to survey interview date were statistically significantly associated with severely stunted children aged 0–23 months or 0–59 months. Another study by Agu et al. [17] further revealed that maternal education, ethnicity and lack of exclusive breastfeeding were significantly related to stunted children aged <2 years in Nigeria. Odunayo and Oyewole [18] carried out a community-based study to examine the risk factors for malnutrition among rural children aged 0–59 months in the Ifewara community of Osun State. They postulated that overcrowding and low maternal income were also related to childhood malnutrition. The limitations of the aforementioned studies were that national estimates could mask the wide disparities in the health and socioeconomic issues of the considered geopolitical zones and/or states, leading to inadequate policy formulation and implementation. In addition, information concerning the changes over time among stunted children across the considered geopolitical zones and/or states was lacking, which was vital because it would assist in the evaluation of the efficacy of previous stunting interventional initiatives and guide the retooling of present or future programmes. Geopolitical zone studies in Nigeria can reveal complex interrelated contextual factors that differentially affect interventional policies on stunting among children below 5 years of age across communities and/or states [19].

To the author’s knowledge, no published population-based studies have examined the trends and factors related to stunting in children in the combined three NGZs (NC, NE and NW) with similar characteristics (e.g., religion, culture and socioeconomic activities). Accordingly, this study examined the trends in the prevalence of stunted children and the associated factors related to stunting, using the Nigeria Demographic and Health Survey (NDHS) dataset for 2008–2018. Estimates from the study will guide health policymakers in formulating cost-effective, zone-specific intervention programmes to reduce stunting in children across the three NGZs in Nigeria.

## 2. Materials and Methods

The NGZs dataset from the NDHS 2008, 2013 and 2018 surveys was extracted and utilised for this study. The respondent’s information concerning health, economic and social wellbeing, such as maternal and child health, anthropometry and demographic data were obtained using identically structured questionnaires. The reported live births from the NGZs population during the surveys were detailed by women aged between 15 and 49 years, who were interviewed during the surveys. Between 2008 and 2018, a weighted total of 62,169 live births of children below 5 years of age occurred in the NGZs (12,760 from the NC, 16,341 from the NE and 33,068 from the NW). Data regarding children aged between 0 and 59 months alive at the time of these surveys and those with valid anthropometric measures were included in the analysis.

In the three combined surveys, 33,682 children below 5 years of age who had valid and complete information regarding date of birth and anthropometric measurements (i.e., height/length in centimetres (cm) in the NGZs were included in the study; of these, 7720, 8823 and 17,139 were from the NC, NE and NW, respectively. The anthropometric measurement procedure used for obtaining the data concerning the height/length of children below 5 years of age prior to the surveys has been detailed elsewhere [5,6,7]. In brief, a Shorr measuring board was used. Children aged between birth and 2 years were measured lying down on the Shorr board (recumbent length), while the height of children older than 2 years was measured standing. To validate the initial measurements, a child was randomly selected from each of the clusters and re-measured a second time, and if the difference was ≤1 cm, then the initial measurements were retained while greater than 1 cm was classified as invalid anthropometric measurements. The children’s height-for-age index was obtained using the WHO’s 2006 child growth reference standard [2].

### 2.1. Study Outcome Variables

The outcome variables were stunting among children aged between 0 and 23 months, 24–59 months and 0–59 months of age. Stunting occurring within the specified age periods was considered binary: each stunted case was determined as the height-for-age Z-score (HAZ) below minus two standard deviations (HAZ < −2SD) coded as ‘1′, and each non-stunted case with HAZ ≥ −2 SD was coded as ‘0′. The determined stunting was guided by the WHO 2006 Child Growth Standard [2].

### 2.2. Potential Associated Independent Factors

The potential confounding factors investigated in the current study were based on the detailed UNICEF nutrition framework [20] and previous studies in low-and middle-income countries [16,17,18,21,22,23,24]. The UNICEF framework consists of direct immediate factors, such as child nutrition and disease occurrence, and past studies have documented their increased association with stunting [25,26,27]. The child nutrition measures included were dietary diversity score and feeding practices (i.e., currently breastfeeding, duration of breastfeeding and breastfeeding initiation), while disease occurrence consisted of infectious disease (such as diarrhoea and fever) in the past 2 weeks prior to the survey interview date and health behaviours (e.g., vaccination). The dietary diversity score was gathered by collating eight food categories consumed within the last 24 h prior to the survey interview. The reported foods consumed were breasted milk, grains root and tubers, legumes and nuts and milk/dairy products. Other foods included flesh foods (meat, fish, poultry and liver/organ meats), vitamin-A rich fruits and vegetables, other fruits and vegetables and eggs; they were divided into two classes (the child had five or more food groups and the child had less than five food groups) [28].

Household wealth index and maternal and paternal education [16,17,29,30,31] have been previously reported to be increasingly associated with stunting; they were grouped as socioeconomic factors. The economic status of the respondents’ households during the surveys was measured using the principal component analysis score [32]. This score-based approach was used to assess the weights of the reported household assets, which included television, radio, refrigerator, car, bicycle, motorcycle, source of drinking water, type of toilet facility, electricity and types of building materials used in the place of dwelling, to estimate the household wealth index factor score. In the three NDHS datasets, the household wealth index factor score was divided into five quintiles (poorest, poorer, middle, richer and richest). However, in the current study, we reclassified the household wealth index into three classes. The top 20% were arbitrarily referred to as rich households, the next 40% represented middle households and the bottom 40% represented poor households.

The results of previous studies have shown that an increased relationship exists between stunting-and individual-related maternal/child characteristics [16,17,21]. The maternal characteristics included in the study were age at birth, height, mother’s body mass index, marital status, birth order and birth interval, while the child’s characteristics included the child’s age and sex. We also considered household decision-related factors, such as healthcare autonomy, movement autonomy and earning/financial autonomy. Listening to the radio, watching television and reading newspapers or magazines were grouped as healthcare knowledge through media. The healthcare service-related factors considered were place of delivery, mode of delivery and delivery assistance. The respondent’s dwelling place during the survey period was used to group the residence type and the geopolitical zone. The residence type and the geopolitical zone were classified as basic or community factors. In all, 29 variables were explored; these aforementioned variables were classified into seven factors: community, socioeconomic, individual (maternal/child), household decision-related, healthcare knowledge through media, healthcare service-related and immediate factors. A detailed categorisation of all these variables is presented in Table 1.

Environmental factors, such as the type of drinking water source, sanitation facility and type of cooking fuel, were not considered because they were part of the household assets used for estimating the economic status of the respondent’s households. Moreover, antenatal and postnatal care services were not included because information concerning them in the surveys was only documented for the most recent birth (or last birth), and the current study was based on all singleton and multiple births in the 5-year period prior to the survey date.

### 2.3. Statistical Analysis

Frequency tabulation of all the possible associated variables highlighted in (Table 1) was carried out for each year of the survey to describe the data utilised in the study, followed by prevalence estimates and their 95% confidence interval (CI) of the study outcomes. The adjusted odds ratios (AORs) for factors associated with the study outcomes were identified by multivariable analyses, which were conducted using multilevel logistics regression. All analyses were carried out using ‘SVY’ command in STATA/MP version 14.1 (StataCorp, College Station, TX, USA) to adjust for clusters and survey weights.

The multivariable logistics regression analyses performed used a stage modelling approach, meaning each of the seven level factors (community, socioeconomic, individual (maternal and child) related, household decision related, healthcare knowledge through media, healthcare service related and immediate factors) was examined each at a time. This approach allows distal factors to be adequately investigated without meddling from proximal factors (e.g., child’s nutrition and disease occurrence referred to as direct or immediate factors). In the first stage modelling, community level factors were entered into the baseline model to examine their strength of association with the study outcomes, followed by a manually stepwise backwards elimination process at 5% significance level. Those variables statistically significant were retained (stage model 1). In the second stage modelling, the socioeconomic level factors were added to the retained significant variables in stage model 1 and again those significantly significant variables were retained (stage model 2). A similar approach was repeatedly utilised for the inclusion of individual maternal and child related, household decision related, healthcare knowledge through media, healthcare service related and immediate factors in the third, fourth, fifth, sixth, and seventh stages, respectively. In the final stage model, the AORs and their 95% CI which showed the variables strength of association with the study outcomes were reported in the study results section.

## 3. Results

During the study period (2008–2018), there were 15,966 stunted children below 5 years of age in the NGZs, of which 5455 were between birth and 23 months of age and 10,511 were between 24 and 59 months of age. The overall prevalence of stunted children (from birth to 59 months of age) was 47.4% (95% confidence interval (CI): 46.4–48.4), while the prevalence of stunted children aged from birth to 23 months was 39.1% (37.9–40.3), and a 53.3% (52.0–54.6) prevalence was observed for stunted children aged between 24 and 59 months (Figure 1).

It was observed that stunting in children aged 24–59 months hardly changed over the 10-year period (Figure 2a). A slightly increasing trend of the prevalence of stunted children in NW was observed from 52.6% (50.4–54.7) in 2008 to 54.9% (52.5–57.2) in 2013 to 56.9% (54.3–59.4) in 2018 (Figure 2b). The prevalence of stunting among children of NC statistically significantly declined from 43.4% (40.6–46.2) in 2008 to 28.9 (26.3–31.6) in 2013, but barely changed between 2013 and 2018.

Over the 10-year period, Jigawa and Kano only experienced a steady increase of stunting in children aged from birth to 59 months, while stunting prevalence in Katsina barely changed during the same period (Figure 3a). In the considered age groups, the prevalence of stunted children slightly decreased among children aged between 6 and 11 months (35.4% in 2008, 34.4% in 2013 and 28.9% in 2018), and a similar pattern was observed for children aged between 48 and 53 months (Figure 3b). Children aged between birth and 5 months in the NGZs reported the lowest prevalence in the three surveys.

The prevalence of stunted children whose mothers had secondary or higher education decreased from 37.6% in 2008 to 28.8% in 2018, but the difference was statistically insignificant. The prevalence of stunting among children from a poor household in NGZs increased by 3.2% from 52.5% in 2008 to 55.7% in 2018. It was further observed that in the three study periods, almost all the independent study factors had a V-shaped trend for the prevalence of stunted children, and the pace of decrease or increase was uneven; the differences were statistically insignificant (Appendix A). Four states (Benue, Kogi, Nassarawa and Niger) in NC zone showed a steady modest decline of stunting in children aged 0–59 months during the study period (Figure 4a). Stunting in children was almost stagnant in all the states in the NE zone except Adamawa and Borno states which decreased from 2008 to 2013 (Figure 4b).

### 3.1. Factors Associated with Stunting among Children Aged between 0 and 23 Months

Appendix A provides detailed findings of all the model analyses of stunted children aged 0–23 months. There were significantly higher odds of stunting among children born to mothers from a poor (AOR = 1.87, 95% CI: 1.39–2.51) or average (AOR = 1.67, 95% CI: 1.27–2.21) household than that in the case of mothers from a rich household. Male children were more likely to be stunted (AOR = 1.68, 95% CI: 1.43–1.99) than their female counterparts, as were children residing in the NE (AOR = 1.91, 95% CI: 1.37–2.64) and NW (AOR = 2.36, 95% CI: 1.79–3.10) geopolitical zones (Table 2). The multivariable findings also revealed that children of mothers having a height of less than 145 cm had a 3.42 times greater odds of stunting, as well as children whose mothers’ perceived their body size at birth as small or very small had a 1.50 times higher likelihood of stunting. It was surprising that children who were fully vaccinated and children who had an adequate dietary diversity score (≥5) were more likely to be stunted.

### 3.2. Factors Associated with Stunting among Children Aged between 24 and 59 Months

Appendix A provides detailed findings of all the model analyses of stunted children aged 24–59 months. Children of mothers who had no schooling (AOR = 1.90, 95% CI: 1.48–2.44) or a low level of education (or primary) (AOR = 1.76, 95% CI: 1.31–2.35) were more likely to be stunted than those of mothers who had secondary or higher education (Table 2). It was also observed that children residing in rural areas (AOR = 1.37, 95% CI: 1.11–1.70) and male children (AOR = 1.22, 95% CI: 1.04–1.43) had a significantly higher probability of stunting (Table 2). Similarly, children who had an episode of diarrhoea in the two weeks prior to the survey interview date (AOR = 1.69, 95% CI: 1.34–2.15), as well as children whose mothers never read newspapers or magazines (AOR = 2.07, 95% CI: 1.11–3.84), were more likely to be stunted. Other significantly higher odds for stunting included the NE and NW geopolitical zones, poor or average households, mother’s height and fourth or higher birth order with a short interval of ≤ 2 years (Table 2).

### 3.3. Factors Associated with Stunting among Children Aged between 0 and 59 Months

Appendix A provides detailed findings of all the model analyses of stunted children aged 0–59 months. Children delivered by traditional birth attendants or relatives at non-health facilities were at a greater odds of stunting (AOR = 1.27, 95% CI: 1.06–1.51), as were children who reported an episode of diarrhoea (AOR = 1.47, 95% CI: 1.25–1.73) two weeks before the survey interview date (Table 2). Furthermore, children of mothers who had no access to television (AOR = 1.32, 95% CI: 1.08–1.61) and children who had a dietary diversity score of at least 5 (AOR = 1.85, 95% CI: 1.48–2.32) were more likely to be stunted. The other associated factors that had a significantly greater likelihood of stunting among children aged 0–59 months were as follows: fourth or higher birth order with a short interval of ≤2 years, poor or average households, mothers who had no schooling or low level of education (or primary), male children, mother’s height and the NE and NW geopolitical zones (Table 2).

## 4. Discussion

The overall prevalence of stunted children across the three age groups examined in the NGZs (NC, NE and NW) was approximately 39% for children aged between birth and 23 months, 53% for children aged between 24 and 59 months and 47% for children aged between birth and 59 months. These observed prevalence values were well above the WHO high prevalence category (30–39%) [11], indicating a serious public health concern in the NGZs. The trends of the prevalence of stunted children over the study period (2008–2018) indicated that there was no statistically significant decrease recorded in the three NGZs except for NC, which reported a sharp decrease between 2008 and 2013. An unequal variation in the prevalence of stunted children across the 19 states in the NGZs was also observed, and only three states from NC (i.e., Benue, Kogi and Niger) reported a modest steady decrease and prevalence well below the WHO standard (<30%) over the study period. This is a distressing and concerning trend, as this could prevent a huge number of children below 5 years of age from reaching their cognitive and physical potential. The prevalence of stunting in the NGZs remains very high, particularly in the NE and NW. As a result, further improvement and urgent interventions are needed to reduce all forms of child malnutrition. 

The study further identified four consistent factors (children residing in the NE and NW, belonging to a poor or average household, being a male child and having a short mother) that were significantly associated with the higher odds of stunting in children across each of the three age groups. Moreover, the increased likelihood of stunting in children aged 24–59 months and 0–59 months was related to having a mother with low or no schooling, fourth or higher birth order with a short birth interval (≤2 years), and a child who had an episode of diarrhoea two weeks prior to the survey date. Furthermore, the dietary diversity score (≥5) was associated with stunting among children aged 0–23 months and 0–59 months with higher odds, while the baby size at birth (small or very small) and having complete vaccination were only associated with stunting in children aged 0–23 months. Living in a rural area and having a mother who never read newspapers or magazines were associated with children aged 24–59 months, and children of mothers who never watched television and children delivered by non-health professionals were related to stunting in children aged 0–59 months.

Children of all three age groups living in the NE or NW had a significantly increased likelihood of stunting compared with those living in the NC. This outcome was consistent with that of earlier studies conducted in Ghana [33], Rwanda [34] and Bangladesh [35], which suggested that the geographical location or region was significantly associated with stunting. However, the current study’s finding was not unexpected because of the NE’s and NW’s previous poor performance records concerning child health indicators in the past two decades; for example, in 2018, the NE ranked second to the NW, with the highest reported under-5 mortality rate in Nigeria [7,36]. A plausible explanation for the current finding may be attributed to the militant insurgency, banditry and cattle rustling in the zones that have cut-off a huge number of farmers and livestock herders from accessing their farmlands for nearly a decade. These issues are devastating because they limit agricultural yield, which may in turn lead to severe food insecurity. This setback could adversely affect the large number of households in the NE and NW, as they largely rely on agricultural activities as their main source of livelihood, particularly those residing in rural areas. The lean season may also contribute to the increased odds of stunting because during this period, food prices are exorbitant up to the harvest season, leading to inadequate availability of nutritional food crops to a vast majority of households. This is supported by a recent study in Nepal, which suggested that children below 5 years of age living in food-insecure households had a higher probability of stunting [37]. Additionally, cultural beliefs and practices are very common in the NE and NW, which may have contributed to the higher odds of stunting. Mwangome et al. [38] and Bhui [39] previously suggested that culture influences health practices by labelling some nutrient-rich food as taboo and traditional herbs given to newborns for at least two weeks of birth to prevent contracting diseases, resulting in depriving children from receiving adequate dietary, immunisation and vitamin A supplementation.

In all age groups, children from poor households or middle-income households had a higher likelihood of stunting than children from rich households. This finding is supported by previous studies carried out in Zambia [40], Iran [41] and Nepal [42]. However, this finding is unsurprising because children from low-income households lack sufficient nutritional food intake, are exposed to poor environmental conditions and have inadequate access to basic healthcare services, which may adversely affect the child’s growth. A plausible reason for the current finding could be linked to a recent report that suggested that approximately 87% of the estimated 83 million poor people in Nigeria live in the NGZs [43]. Each day in Nigeria, these poor people live below USD 1 (or NGN 415) [44]. This economic limit affects poor households in several ways, including affordable nutritional food, living in an environment with good water and sanitation infrastructure, and improved childcare practices (i.e., immunisation, preventative care and suitable and timely feeding), resulting in a greater likelihood of stunting in children.

The increased odds of the relationship between maternal height and stunting among children of all age groups observed in the study was evidently the highest for maternal height of less than 145 cm. This finding is in agreement with a previous study that used the combined DHS dataset of 54 low-and middle-income countries, which suggested an inverse relationship; that is, a 1-cm increase in maternal height was associated with a decreased odds of stunting in children aged 0–59 months [29]. Other similar studies conducted in India [45] and Bangladesh [46] also indicated an increased likelihood of stunting among children with short mothers. The elevated odds of stunting related to maternal height could be attributed to the maternal organ size (i.e., pelvic), maternal poor nutrition status and foetal programming, which adversely impact the unborn child’s growth *in utero*. It has been previously suggested that short women have an increased probability of having a narrow pelvis, which impacts the uterus for optimum foetal growth, resulting in low birth weight (LBW) or small-for gestational-age babies [47,48]. Babies born under this condition are more prone to infectious diseases, have increased odds of acute undernutrition and have impaired absorption of vital micronutrients [49], leading to early child stunting. The significantly higher odds of association between LBW (or children perceived as small or very small after birth by their mothers used in place of the actual birth weight as a proxy) and stunting among children aged 0–23 months observed could be attributed to the aforementioned adverse effects of maternal organ size, maternal poor nutrition status and foetal programming.

Male children had a significantly increased odds of association with stunting consistently across each of the three age ranges compared with their female counterparts. This outcome is consistent with that of a similar stunting study conducted in Madagascar [50], Ghana [33] and Indonesia [51]. The cause of this gender difference in stunting remains unclear, and these studies have struggled to find a uniform explanatory factor or conjecture a plausible reason. However, it could be possible that because of cultural beliefs in sub-Saharan Africa, including Nigeria, female children are given more nutritional food for rapid growth and maturity than their male peers in order to gain a higher bride dowry [52]. A previous study in Bangladesh suggested that male children spend more time outdoors than their female peers [53], leading to male children being exposed to and vulnerable to environmental pollution and health hazard contaminants [54]. This may affect their growth as their body organs (i.e., immune, respiratory and digestive systems) are still developing [55].

A higher likelihood of stunting among children aged 24–59 months and 0–59 months was associated with mothers who had no schooling or a low level of (or primary) education compared with those who had secondary or higher education. This finding is in line with that of earlier studies conducted in Bangladesh [56], Kenya [57] and Pakistan [58]. It is possible that little-educated or uneducated mothers are less likely to comprehend the full benefits of good childcare practices, such as hygienic behaviours, immunisation, preventative care and suitable and timely feeding, and this may have increased the odds of the stunting observed. It has been previously revealed that a remarkable number of women do not promptly seek care because of an inadequate understanding of the benefits that enhance child survival [59]. Additionally, little-educated or uneducated mothers are more likely to strictly adhere to socio-cultural practices that may negatively affect the child’s health (e.g., cultural prelacteal feeding practices that deprive newborns of colostrum or the first breast fluid that is rich in nutrients and immunoglobulins) [60]. Therefore, empowering women through education is a crucial pathway that could substantially reduce stunting in Nigeria, particularly in the NGZs where culture still discriminates against women’s education. Educated women are more likely to support their children with timely healthcare, better child rearing and delayed childbearing [61]. It will also give them leverage to fend for themselves and their families, elevate their self-confidence and influence health resources [62]. Children aged 24–59 months who lived in rural areas were more likely to be stunted than those of their urban peers. This finding is in contrast to that of previous studies [63,64]; however, other similar studies have shown a significant association with stunting [34,65]. A plausible explanation of the current finding could be linked to inadequate infrastructural development (i.e., improved water supply and basic sanitation facilities) [66] and environmental pollution, such as that emanating from solid fuel usage; previous studies [67] have shown that unimproved water source and sanitation and solid fuel pollution are associated with deaths of children below 5 years of age. Fink et al. [68] and Agho et al. [34] also suggested in their studies that an unimproved source of drinking water was strongly associated with severe stunting. Children crawl and walk around actively with increased exposure to environmental contaminants during this period, thereby contracting infectious disease (i.e., diarrhoea).

Children aged 0–23 months and 0–59 months who had acceptable minimum dietary diverse food intake (≥5) 24 h prior to the survey interview were more likely to be stunted than their counterparts who had inadequate dietary diversity (<5). It has been suggested by earlier studies [21,69] that the consumption of a minimum acceptable diet with dietary diversity and an adequate intake of iron-rich foods reduces the odds of stunting. The contradiction observed in the current study remains unclear. However, it is possible that the timely initiation of complementary feeding and the consumption of other unhealthy complementary foods along with diverse dietary foods contributed to the unexpected higher odds for adequate dietary diversity food. The odds of stunting among children aged 24–59 and 0–59 months were significantly greater for children who had diarrhoea 2 weeks prior to the survey interview date than those who did not report a diarrhoea episode. This finding is inconsistent with a similar study conducted in Nepal, which indicated a statistically insignificant relationship between a diarrhoea incident two weeks before the survey date and childhood stunting [70]. Nevertheless, other studies have shown a significant association with stunting [35,71,72]. The possible reason for the current finding was inadequate dietary intake, which often leads to poor nutritional status, resulting in reduced appetite, increased catabolism and impaired intestinal absorption, which adversely impact growth [3].

Children aged 0–59 months delivered by non-health professionals (i.e., traditional birth attendants (TBAs), relatives, or friends) had increased odds of being stunted compared with those delivered by health professionals such as doctors and nurses/midwives. This finding is not in line with that of a recent study conducted in five South Asian countries by Wali et al. [21], which found that children delivered by TBAs were less likely to be stunted than those delivered by doctors or nurses. This difference may be linked to the fact that in some South Asian countries, TBAs are trained and integrated into healthcare systems. Such training includes supporting nursing mothers on appropriate and timely breastfeeding and complementary feeding and providing advice on primary healthcare services related to mothers, particularly inexperienced mothers [73,74,75]. Furthermore, children of birth order 4 or higher who were born with a shorter birth interval of ≤2 years had a higher probability of stunting than those with a longer birth interval of >2 years, particularly for children aged 24–59 months of age. The higher odds of stunting observed could be attributed to the shorter gaps between births and pregnancies, which often lead to inattention and improper care for higher-order children’s needs. This can adversely affect a child’s health (e.g., wasting, stunting and being underweight), and previous studies have suggested that at least three years longer child spacing is more likely to protect against child stunting [76,77]. It was also observed in the study that the increased likelihood of stunting was associated with children whose mothers never read newspapers or magazines for children aged 24–59 months, while children whose mothers never watched television were more likely to be stunted than those whose mothers watched television, particularly in the case of children aged 0–59 months. Mass media exposure (electronic or print) remains a crucial source of health information, as it disseminates health-related programmes, such as the benefits of breastfeeding, immunisation and complementary feeding. Furthermore, children aged 0–23 months and 0–59 months who were breastfed for at least 12 months were less likely to be stunted than those breastfed for less than 12 months. This finding is in contrast to that of previous studies conducted in Nigeria [16] and Nepal [42].

The limitations of the current study were as follows: (1) The real income of households, which remains an important factor influencing stunting among children, was not examined in the study because it was lacking in the NDHS dataset. However, the household asset-based score estimates used in place of actual household income may have affected the stunting odds estimates reported. (2) Child stunting odds estimates may have been underestimated or overestimated because of the potential confounding residual from the medical condition of mothers (e.g., infection, diabetes, hypertension and maternal depression after childbirth) and children (e.g., birth asphyxia, jaundice and sepsis) were not adjusted for because of the unavailability of data. (3) Other important immediate or direct causes of child malnutrition, such as unhealthy food, inappropriate child feeding practices and household food insecurity, were also not included in the study because of the unavailability of data. (4) A causal effect could not be measured because the analyses used retrospective cross-sectional data to identify factors related to stunting in children. (5) Data concerning a child’s dietary arrangements was lacking in the NDHS dataset as household food insecurity may result in the rationing and skipping of meals, and previously, it was suggested that these changes are associated with households with severe food insecurity [78].

Despite the highlighted limitations, this study observed some strengths: (1) The restriction of the analyses to births within a five-year period prior to the surveys decreased the potential recall errors (i.e., from the child’s birth date) and measurement errors (i.e., from the child’s anthropometric measurement). (2) The findings obtained may reflect the true situation of a child’s stunting in the NGZs because the studied population had similar characteristics (e.g., religion, culture and socioeconomic activities), which will aid an effective formulation of intervention initiatives. (3) The child stunting estimates observed in this study were population based, which increases the validity and can be generalised to regions or countries with similar settings. (4) Statistical differences were well identified because of the considerable statistical power of the combination of the three NDHS datasets.

## 5. Conclusions

The findings showed that four common factors, including children residing in the NE or the NW, residing in a poor or average-income household, being a male child and children of short mothers, were significantly associated with an increased likelihood of stunting in children across each of the three age categories (0–23 months, 24–59 months and 0–59 months) in the NGZs, Nigeria. Other significant factors that increased the odds of stunting in children, particularly those aged 24–59 months and 0–59 months, were low maternal education or no schooling, diarrhoea and fourth or higher birth order with a short birth interval of ≤2 years, while children perceived to be small or smaller by their mothers at birth had higher odds of stunting among children aged 0–23 months. Furthermore, children living in rural areas and children whose mothers had no access to electronic media (i.e., television) or print media (i.e., newspapers or magazines) had an increased likelihood to be stunted. These findings suggest an urgent need for interventions at both the community and individual levels. At the community level, interventions should focus on the provision of social safety nets (e.g., cash transfer, retooling of school feeding programs and special relief during the lean season or drought) and the provision of clean water sources and sewage systems to reduce the incidence of diarrhoea in the community. The coverage of these interventions should be expanded in areas with limited social and economic support (i.e., rural communities and urban slum areas), and such interventions should aim at low socioeconomic households. Interventions at the individual level should focus on sensitising and educating poor mothers on the benefits of gender-balanced initiation of supplementary feeding, timely monitoring of the offspring of a short mother, family planning services (e.g., child spacing), skin-to-skin care of small babies and personal hygiene. These intervention initiatives will substantially reduce the prevalence of stunting in children if implemented across the three NGZs in Nigeria.

## Figures and Tables

**Figure 1 nutrients-13-04312-f001:**
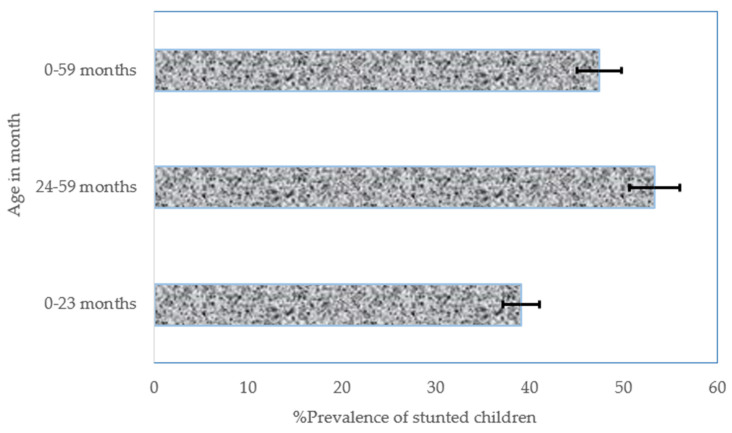
The overall prevalence of stunted children between 2008 and 2018 in the NGZs by age in months. Notes: If 95% confidence interval (CI) bars do not overlap, it implies that the difference is statistically significant (*p*–value < 0.05).

**Figure 2 nutrients-13-04312-f002:**
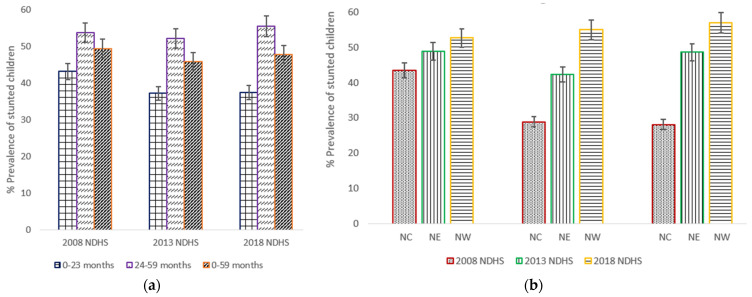
(**a**) Trends in prevalence of stunted children aged 0–23 months, 24–59 months, and 0–59 months, with 95% confidence interval by year of the northern geopolitical zones (NGZs) of Nigeria Demographic and Health Surveys (NDHS), 2008–2018; (**b**) Trends in prevalence of stunted children aged between 0 and 59 months by the Northern geopolitical zones (north central (NC), northeast (NE) and northwest (NW)). Notes: If 95% confidence interval (CI) bars do not overlap, it implies that the difference is statistically significant (*p*–value < 0.05).

**Figure 3 nutrients-13-04312-f003:**
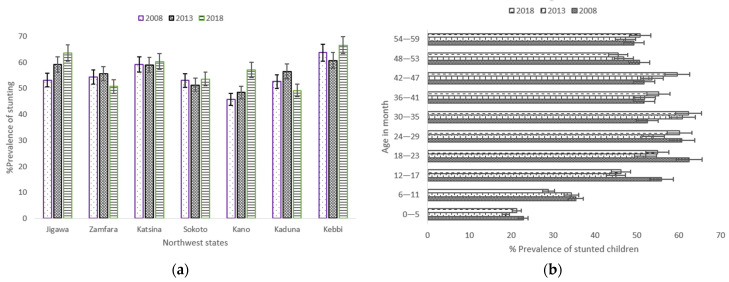
(**a**) Trends in prevalence of stunted children aged 0–59 months, with 95% confidence interval by Northwest states 2008–2018; (**b**) Trends in prevalence of stunted children aged between 0–59 months by child’s age. Notes: If 95% confidence interval (CI) bars do not overlap, it implies that the difference is statistically significant (*p*–value < 0.05).

**Figure 4 nutrients-13-04312-f004:**
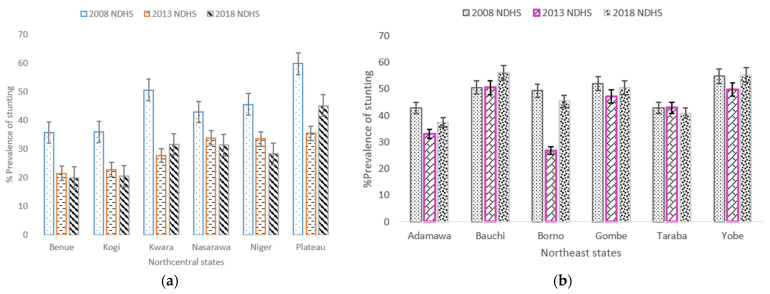
(**a**) Trends in prevalence of stunted children aged 0–59 months, with 95% confidence interval by Northcentral states 2008–2018; (**b**) Trends in prevalence of stunted children aged 0–59 months by Northeast states. Notes: If 95% confidence interval (CI) bars do not overlap, it implies that the difference is statistically significant (*p*–value < 0.05).

**Table 1 nutrients-13-04312-t001:** Definition and classification of potential confounding factors related to child stunting for the study analysis.

Independent Variable	Classification
Community—Level factor	
Residence	1 = urban; 2 = rural
Geopolitical zone	1 = North Central; 2 = Northeast; 3 = Northwest
Socioeconomic factor	
Household wealth index	1 = Poor; 2 = Middle; 3 = Rich
Maternal educational level	1 = No education; 2 = Primary; 3 = Secondary or higher
Maternal occupation	1 = Not working; 2 = Working
Father’s educational level	1 = No education; 2 = Primary; 3 = Secondary or higher
Number of women in the Household	1 = One woman; 2 = At least 2 women
Individual–related factor (Maternal and child)	
Mother’s age at child birth (years)	1 = < 20; 2 = 20–29; 3 = 30–39; 4 = 40–49
Mother’s body mass index (kg/m^2^) (MBMI)	1 = Normal (18.5 ≤ MBMI ≤ 24.9); 2 = Underweight(MBMI < 18.5); 3 = Overweight (25 ≤ MBMI ≤ 29.9); 4 = Obese (MBMI ≥ 30)
Contraceptive use	Yes used any form of contraceptive; No otherwise
Maternal height (centimeter (cm))	1 = ≥ 160; 2 = 155–159; 3 = 150–154; 4 = 145–149; 5 = < 145
Birth order and birth interval	1 = second or third child, interval > 2 years; 2 = First child; 3 = second or third child, interval < = 2 years; 4 = fourth or higher child, interval > 2 years; 5 = fourth or higher child, interval < = 2 years
Child’s sex	1 = Female; 2 = Male
Mother’s perceived baby size at birth	1 = Average or larger; 2 = Small or very small
Child’s age	1 = 0–5; 2 = 6–11; 3 = 12–17; 4 = 18–23
Health knowledge through (media exposure)	
Frequency of listening to radio	1 = At least once a week; 2 = Less than once a week; 3 = Never
Frequency of watching television	1 = At least once a week; 2 = Less than once a week; 3 = Never
Frequency of reading newspaper or magazine	1 = At least once a week; 2 = Less than once a week; 3 = Never
Influence over household decision making	
Woman has money autonomy	1 = By husband/partner alone or someone else; 2 = woman alone or joint decision
Woman has healthcare autonomy	1 = By husband/partner alone or someone else; 2 = woman alone or joint decision
Woman has movement autonomy	1 = By husband/partner alone or someone else; 2 = woman alone or joint decision
Health service related factor	
Place of birth	1 = Home; 2 = Health facility
Mode of delivery	1 = non-caesarean; 2 = caesarean section
Delivery assistance	1 = Health professional; 2 = non-Health professional
Immediate-level factor	
Dietary diversity score	1 = < 4 foods/inadequate; 2 = ≥ 4 foods/adequate
Initiation of breastfeeding	1 = Within 1 h of birth; 2 = more than 1 h after birth, refers to as delayed breastfeeding initiation
Currently breastfeeding	1 = yes currently breastfeeding; 2 = no otherwise
Duration of breastfeeding	1 = up to 12 months; 2 = more than 12 months
Vitamin A supplement	Yes if the child received vitamin A supplementation; no otherwise
Full vaccination	Yes if the child received a Bacillus Calmette–Guérin vaccination against tuberculosis; 3 doses of Diptheria, pertussis, and tetanus vaccine; ≥3 doses of polio vaccine; and 1 dose of measles vaccine; no otherwise
Had diarrhea in the last 2 weeks	Yes if child had diarrhea; no otherwise
Had fever in the last 2 weeks	Yes if child had fever; no otherwise

kg, weight measured in kilograms; m^2^, height measured in square meters.

**Table 2 nutrients-13-04312-t002:** Adjusted ORs (95% CI) for independent variables significantly associated with stunted children aged 0–23 m, 24–59 m, and 0–59 m in NGZs, Nigeria (2008–2018).

	Stunted Child (0–23 m)	Stunted Child (24–59 m)	Stunted Child (0–59 m)
Variable	OR ‡ (95% CI)	OR ‡ (95% CI)	OR ‡ (95% CI)
Community level factor			
Residence type			
Urban	-	Ref	-
Rural	-	1.37 (1.11–1.70)	-
Geopolitical zones (North)			
North Central	Ref	Ref	Ref
North East	1.91 (1.37–2.64)	1.96 (1.55–2.48)	1.74 (1.40–2.15)
North West	2.36 (1.79–3.10)	3.23 (2.58–4.04)	2.48 (2.04–3.03)
Socioeconomic factor			
Household wealth index			
Rich	Ref	Ref	Ref
Middle	1.67 (1.27–2.21)	1.55 (1.24–1.95)	1.52 (1.26–1.84)
Poor	1.87 (1.39–2.51)	1.49 (1.15–1.91)	1.52 (1.23–1.88)
Mother’s education			
Secondary or higher	-	Ref	Ref
Primary	-	1.76 (1.31–2.35)	1.43 (1.15–1.77)
No education	-	1.90 (1.48–2.44)	1.54 (1.28–1.84)
Mother’s working status			
Not working	-	-	-
Working	-	-	-
Father’s education			
Secondary or higher	-	-	-
Primary	-	-	-
No education	-	-	-
Number of women in household			
One woman	-	-	-
At least 2 women	-	-	-
Individual level factor (maternal)			
Mother’s age (years)			
<20	-	-	-
20–29	-	-	-
30–39	-	-	-
40–49	-	-	-
Mother’s body mass index (kg/m^2^) (MBMI)			
Underweight (MBMI < 18.5)	-	-	-
Normal (18.5 ≤ MBMI ≤ 24.9)	-	-	-
Overweight or Obese (25 ≤ MBMI ≤ 29.9)/(MBMI ≥ 30)	-	-	-
Birth order/birth interval			
First	-	1.01 (0.80–1.28)	1.16 (0.97–1.38)
2nd or 3rd rank, interval ≤ 2 years	-	1.32 (0.98–1.79)	1.30 (1.02–1.66)
2nd or 3rd rank, interval > 2 years	-	Ref	Ref
4th or higher rank, interval > 2 years	-	1.01 (0.81–1.26)	1.09 (0.94–1.29)
4th or higher rank, interval ≤ 2 years	-	1.46 (1.13–1.88)	1.53 (1.24–1.89)
Contraceptive use			
Yes	-	-	-
No	-	-	-
Maternal height (centimeter (CM))			
≥160	Ref	Ref	Ref
155–159	1.35 (1.06–1.72)	1.93 (1.60–2.32)	1.66 (1.43–1.93)
150–154	1.53 (1.20–1.96)	2.01 (1.62–2.49)	1.78 (1.52–2.09)
145–149	1.97 (1.38–2.80)	2.39 (1.74–3.29)	2.18 (1.72–2.77)
<145	3.42 (1.70–6.87)	2.86 (1.24–6.59)	4.05 (2.24–7.31)
Sex of child			
Female	Ref	Ref	Ref
Male	1.68 (1.43–1.99)	1.22 (1.04–1.43)	1.37 (1.21–1.55)
Mother’s perceived baby size			
Average or larger	Ref	-	-
Small or very small	1.50 (1.18–1.91)	-	-
Health knowledge through (media exposure)			
Frequency of listening to radio			
At least once a week	-	-	-
Less than once a week	-	-	-
Never	-	-	-
Frequency of reading newspaper or magazine			
At least once a week	-	Ref	-
Less than once a week	-	2.02 (0.98–4.14)	-
Never	-	2.07 (1.11–3.84)	-
Frequency of watching television			
At least once a week	-	-	Ref
Less than once a week	-	-	1.04 (0.81–1.32)
Never	-	-	1.32 (1.08–1.61)
Influence over household decision making			
Woman has earning autonomy			
By husband/partner alone or someone else	-	-	-
woman alone or joint decision	-	-	-
Woman has healthcare autonomy			
By husband/partner alone or someone else	-	-	-
woman alone or joint decision	-	-	-
Woman has movement autonomy			
By husband/partner alone or someone else	-	-	-
woman alone or joint decision	-	-	-
Health service related factor			
Place of birth			
Health facility	-	-	-
Home	-	-	-
Mode of delivery			
Non-caesarean			
Caesarean	-	-	-
Delivery assistance			
Health professional	-	-	Ref
Non-health professional			1.27 (1.06–1.51)
Immediate related factor	-	-	-
Dietary diversity score ^			
<5 foods/inadequate	Ref	-	Ref
≥5 foods/adequate	1.92 (1.55–2.37)	-	1.85 (1.48–2.32)
Initiation of breastfeeding ^			
More than 1 h after birth	-	-	-
Within 1 h of birth	-	-	-
Currently breastfeeding ^			
No	-	-	-
Yes	-	-	-
Duration of breastfeeding ^			
up to 12 months	Ref	-	-
more than 12 months	0.47 (0.26–0.83)	-	-
Full vaccination			
No	Ref	-	-
Yes	1.30 (1.03–1.64)	-	-
Had diarrhea in the last 2 weeks			
No	-	Ref	Ref
Yes	-	1.69 (1.34–2.15)	1.47 (1.25–1.73)
Had fever in the last 2 weeks			
No	-	-	-
Yes	-	-	-

Notes: OR (95%CI), Odds ratios with corresponding 95% confidence interval; m, months; Ref, reference category; NGZs, three northern geopolitical zones (northcentral, northeast and northwest); kg, weight measured in kilograms; m^2^, height measured in square meters; ‡, adjusted independent variables, residence type, region, household wealth status, maternal education, maternal work status, paternal education, number of women in the household, mother’s age, MBMI, contraceptive use, maternal height, perceived baby size by their mothers, child sex, birth order/interval, listening to radio, reading newspaper or magazine and watching television, power over earning, autonomy over healthcare, purchasing decision, delivery assistant, mode of delivery, place of delivery, dietary diversity, early initiation of breastfeeding, currently breastfeeding, duration of breastfeeding, vaccination, diarrhea in the last two weeks and fever in the last two weeks; ^, independent variables that were not adjusted for children aged 24–59 months.

## Data Availability

This study was based on a public domain dataset that is freely available online: https://dhsprogram.com/data/dataset/Nigeria_Standard-DHS_2018.cfm?flag=0; https://dhsprogram.com/data/dataset/Nigeria_Standard-DHS_2013.cfm?flag=1 and https://dhsprogram.com/data/dataset/Nigeria_Standard-DHS_2008.cfm?flag=1 (accessed on 8 May 2020).

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
