# Peer review of "Trends of Stunting Prevalence and Its Associated Factors among Nigerian Children Aged 0–59 Months Residing in the Northern Nigeria, 2008–2018"

_nutrients, 2021, doi:10.3390/nu13124312_

Round 1

Reviewer 1 Report

This was an enormous study of the risk factors for stunting in northern Nigeria.  The methodology, analysis, and presentation were excellent.  I have only a couple suggestions.

Page 8, Figure 2b.  Something is wrong here.  Is each of these bars a year or a province?  There are too many bars and not enough labels.  In addition, this figure would benefit from the colors used in the other figures.

Tables 3and 4.  All 3 tables consume 15 pages of the manuscript.  They are highly repetitive.  I suggest removing Table 3 and 4 and simply discussing in the text any major differences between the age groups.  The Discussion spent considerable time stressing the common features of the age groups rather than their differences.  This supports this suggestion of removing the two large tables.

Reviewer 2 Report

This manuscript examined the trends in the prevalence of stunting in the northern Nigeria and identified risk factors, which may inform public heath policy making.

Line 60: What was the reason for less prioritizing stunting? Was the prevalence of wasting decreased? This sentence also seems to contradict to the next paragraph where multiple programs that focused on reducing the burden of child malnutrition (possibly including stunting) are introduced. I’m just not sure whether the high prevalence of stunting is attributable to low priority of stunting among various intervention goals.

Line 121: How were participants sampled? Are data representative of NGZs population?

Line 132: What if the difference was bigger than 1 cm?

Line 191: Did NDHS only recruit singleton births? Or did the authors exclude births other than singleton births? Please clarify.

Line 200: Please clarify what the authors meant to describe with ‘multilevel logistic regression.’ Did the authors refer to the models that account for clustering of subjects? (https://www.ncbi.nlm.nih.gov/pmc/articles/PMC5575471/) I couldn’t find any details on multilevel logistic regression models.

Line 231: How were trends tested statistically? I don’t see any descriptions on statistical significance of trends in the Methods section. Also, please present statistical significance on each Figure with appropriate footnotes.

Figure 2. Please make it clear that (a) is among the Northern geopolitical zones of Nigeria, not the entire Nigeria

Discussion: The authors’ efforts to explain the results are mostly insightful but are sometimes speculative. Please review the discussion and streamline speculative hypotheses and less scientific explanations.

Line 551: Are the prevalence estimates presented in Figures unadjusted? If not, please describe how the prevalence estimates were adjusted. If yes, please clarify this sentence.

Conclusion: Given the cross-sectional nature of this analysis, please tone down the policy suggestions.